# Pyrazinoic Acid Inhibits the Bifunctional Enzyme (Rv2783) in *Mycobacterium tuberculosis* by Competing with tmRNA

**DOI:** 10.3390/pathogens8040230

**Published:** 2019-11-12

**Authors:** Lei He, Peng Cui, Wanliang Shi, Qiong Li, Wenhong Zhang, Min Li, Ying Zhang

**Affiliations:** 1Department of Laboratory Medicine, Renji Hospital, School of Medicine, Shanghai Jiaotong University, Shanghai 200127, China; buningweishi_1985@126.com (L.H.);; 2Department of Infectious Diseases, Huashan Hospital, Fudan University, Shanghai 200040, China; 3Department of Molecular Microbiology and Immunology, Bloomberg School of Public Health, Johns Hopkins University, Baltimore, MD 21205, USA

**Keywords:** mycobacterium tuberculosis, pyrazinamide resistance, *Rv2783c* gene, PNPase, tmRNA

## Abstract

Pyrazinamide (PZA) is a key drug for tuberculosis treatment. The active form of PZA, pyrazinoic acid (POA), appears to inhibit multiple targets in *M. tuberculosis.* Recently, the bifunctional enzyme Rv2783 was reported as a new target of POA. However, the mechanism by which POA inhibits Rv2783 is not yet clear. Here, we report how a new A2104C substitution in *Rv2783c*, identified in PZA-resistant clinical isolates, conferred resistance to PZA in *M. tuberculosis*. Expression of the mutant allele recapitulated the PZA resistance. All catalytic activities of Rv2783, but not the mutant, were inhibited by POA. Additionally, POA competed with transfer-messenger RNA (tmRNA) for binding to Rv2783, other than the mutant. These results provide new insight into the molecular mechanism of the antitubercular activity of PZA.

## 1. Introduction

Pyrazinamide (PZA) is a critical front-line tuberculosis (TB) drug that plays a key role in shortening TB therapy from 9–12 months to 6 months [1,2] due to its unique activity against persisters, which are not killed by other first-line TB drugs [3,4]. PZA is a prodrug that is required to be converted into its active form, pyrazinoic acid (POA), by pyrazinamidase (PZase) encoded by *pncA* in *M. tuberculosis* (*Mtb*) [5]. The ribosomal protein S1 (RpsA, Rv1630), a crucial protein involved in both regular translation and the ribosome-rescuing trans-translation process via combining with transfer-messenger RNA (tmRNA), has been reported as a target of POA [6]. Further structural studies indicate that POA binds to RpsA through hydrophobic interactions by hydrogen bonds, which are mainly mediated by residues (Lys303, Phe307, Phe310, and Arg357) that are also crucial for tmRNA binding [7]. *rpsA* mutations seem to play a role in the PZA resistance of some other clinical isolates without *pncA* mutations [8,9]. However, *rpsA* polymorphisms are seemingly found in some PZA-susceptible clinical strains from closely related geographical regions. Additionally, there are some low-level PZA-resistant clinical strains (minimal inhibitory concentration (MIC) = 200–300 µg/mL, pH 6.0) without *pncA* and *rpsA* mutations [8,10,11,12,13,14,15]. Subsequently, mutations in *panD* encoding aspartate decarboxylase were suggested to be an additional mechanism of PZA resistance, and the PanD protein involved in β-alanine synthesis was also reported as a novel target of POA/PZA [16,17,18].

Recently, a bifunctional enzyme, guanosine pentaphosphate synthetase (GpsI)/polyribonucleotide nucleotidyltransferase (PNPase) (Rv2783), involved in RNA degradation, which was originally identified as a protein that binds POA [6], was found to be associated with PZA resistance and was identified as another target of POA in *M. tuberculosis* [19]. However, how POA inhibits Rv2783 and how mutations in *Rv2783c* cause PZA resistance are not yet known. Herein, we found a new mutant site (A2104C) of the *Rv2783c* gene in PZA-resistant *M. tuberculosis* isolates, which is located in the S1_PNPase domain including the KXXXFXXF motif that also exists in RpsA and is essential for binding both POA and tmRNA. We demonstrated that both Rv2783 and the Rv2783_K702Q_ mutant protein (encoded by the A2104C mutation) could degrade Poly(A), and that POA competes with tmRNA for binding to Rv2783, but not Rv2783_K702Q_, due to the A2104C mutation, which hampers POA binding.

## 2. Results

### 2.1. The M. tuberculosis Rv2783c_A2104C_ Mutant is Associated with PZA Resistance

Based on our previous report showing that selected proteins encoded by *Rv2731*, *Rv2783c*, *Rv3169*, and *Rv3601c* could bind to a POA derivative, 5-hydroxyl-2-pyrazinecarboxylic acid [6], and another report demonstrating Rv2783 as a possible target of POA [19], we performed Sanger sequencing of the target gene Rv2783c from a panel of 56 PZA-resistant clinical isolates of *M. tuberculosis* and identified the same A2104C mutation in *Rv2783c* in two clinical strains. In addition, we found no *Rv2783c* mutations in 42 PZA-susceptible extensively drug-resistant tuberculosis (XDR-TB) clinical isolates without *pncA* and *rpsA* mutations (Table 1).

The two PZA-resistant clinical isolates had no *pncA*, *rpsA*, or *panD* mutations (Table 1), which are associated with PZA resistance. The PZA MICs of the two clinical isolate strains with the A2104C mutation in Rv2783c were 200–300 μg/mL compared to the 100 μg/mL in the sensitive control strain *M. tuberculosis* H37Rv. PZase assay also showed that the two mutants were positive for PZase activity, ruling out the possibility of alterations in the promoter or potential regulatory regions of *pncA* that could result in a lack of PZase activity as a possible cause of the PZA resistance in the two mutants. The above findings suggest a potential new mechanism of PZA resistance in the two PZA-resistant mutants, other than the known *pncA*, *rpsA*, and *panD* mutations.

### 2.2. Expression of Rv2783cA2104C in M. tuberculosis H37Ra Causes PZA Resistance

We further explored the potential role of *Rv2783c* in PZA susceptibility via engineering a recombinant *M. tuberculosis* strain to express the *Rv2783c* gene with the A2104C mutation (*Rv2783c_A2104C_*) and comparing its PZA resistance to that of wild-type *M. tuberculosis* (Table 2). The recombinant *M. tuberculosis* strain that expressed the *Rv2783c_A2104C_* mutant in *M. tuberculosis* H37Ra using the pOLYG vector caused a six-fold increase in the minimal inhibitory concentration (MIC) of PZA (MIC = 600 μg/mL) compared to the MICs for the empty pOLYG vector control and the parental *M. tuberculosis* H37Ra strain (MICs = 100 μg/mL) at pH 5.5. However, expression of the wild-type *Rv2783c* gene in the same vector in the recombinant *M. tuberculosis* H37Ra strain caused only a two-fold increase in the MIC, i.e., 200 μg/mL PZA. Introduction of the K702Q point mutation into the *M. tuberculosis* H37Ra strain did not alter the susceptibility to other drugs including isoniazid (INH) and kanamycin (Kan).

To further confirm the expression of the *Rv2783c* gene, we compared the production of *Rv2783c* mRNA in the *M. tuberculosis* H37Ra with the empty pOLYG vector and in the recombinant strains with the expressed wild-type or mutant *Rv2783c* gene to the parental H37Ra strain by quantitative reversed transcriptase PCR (qRT-PCR). The *Rv2783c* mRNA level in the strain with the empty pOLYG vector exhibited no significant change compared to the parental *M. tuberculosis* H37Ra strain. On the other hand, the level in each strain with recombinant plasmids showed a nearly 150-fold increase, irrespective of whether the wild-type or mutant *Rv2783c* sequence was carried on the plasmid (Figure 1). Hence, the point mutation (A2104C) in *Mtb Rv2783c* discovered from PZA-resistant clinical strains was identified to confer a higher PZA resistance level without any significant change in the expression level of *Rv2783c*, demonstrating that the mutated site (A2104C) itself is associated with PZA resistance.

### 2.3. Rv2783_K702Q_ Mutant Protein Retains PNPase Activity But Loses Ability to Bind POA

Rv2783 is predicted to function as a polynucleotide phosphorylase (PNPase) enzyme, which plays a direct role in poly(A) degradation. PNPase degrades poly(A) using Pi to generate ADP [20]. The ADP generated from the PNPase can be converted to ATP using an ATP-generating enzyme, such as a kinase with a phosphate group donor substrate. The ATP generated can then be consumed by an ATP-dependent enzyme, such as a luciferase, to generate light. The ADP-Glo™ Kinase Assay (Promega) can be used to monitor the activity of virtually any ADP-generating enzyme using up to 1 mM ATP. Thus, it was considerably relevant to determine whether Rv2783 could catalyze poly(A) degradative activities (Figure 2a). Both the Rv2783 and Rv2783_K702Q_ proteins catalyzed the degradation to the same extent (Figure 2b,c). However, we observed that the enzymatic activity of the wild-type Rv2783, but not mutant Rv2783_K702Q_, was significantly inhibited by POA in a concentration-dependent manner (Figure 2d,e, Table 3). The above results imply that the mutant Rv2783_K702Q_ protein, but not the wild-type Rv2783 protein, could withstand the repressive effects of POA.

### 2.4. Domain Alignment of Rv2783 to Predict Possible Essential Sites for POA and tmRNA Binding

The structure of RpsA (Rv1630) has been solved [7] and it contains four homologous RNA-binding domains, represented as R1 to R4 (Figure 3b). It is revealed by structure analysis that POA binds to Rv1630 by hydrogen bonds and hydrophobic interactions, regulated mainly by residues (Lys303, Phe307, Phe310, and Arg357) in the R4 domain. The first three residues comprising the motif of KXXXFXXF are essential for tmRNA binding [7]. We blasted the conserved domains of Rv2783 and found that Rv2783 also has the S1_PNPase domain (Figure 3a). We then aligned the amino acid sequence of Rv2783 with the four domains of Rv1630 and detected that the S1_PNPase domain of Rv2783 also includes the motif of KXXXFXXF (Lys675, Phe679, Phe682) (Figure 3b). Coincidentally, the K702Q amino acid mutation that conferred PZA resistance is also located in the S1_PNPase domain. Therefore, we inferred that this site might also be an essential site for the competition between POA and tmRNA to bind to Rv2783.

### 2.5. POA Competes with tmRNA to Bind Wild-Type Rv2783 and Not Rv2783K702Q Mutant Protein

We then assessed whether the mutant Rv2783 has any deficiency in the ability of tmRNA binding compared to the wild-type *Mtb* Rv2783, and further estimated whether the interplay between Rv2783 and tmRNA would be influenced by POA. The specific binding of Rv2783 to tmRNA was evaluated via changes in gel mobility with the presence of redundant tmRNA. Both the wild-type *Mtb* Rv2783 and mutant Rv2783_K702Q_ bound to the tmRNA in the absence of POA in a concentration-dependent manner (Appendix A), although the mutant appeared to bind less well overall (Figure 4a). On the other hand, when the system was added with POA, wild-type *Mtb* Rv2783 was repressed from binding to tmRNA at the MIC concentration of POA (1.25 µg/mL) in a concentration-dependent manner, while the binding ability of the mutant Rv2783_K702Q_ was not hampered (Figure 4b).

## 3. Discussion

Herein, we addressed the role of the *M. tuberculosis* gene *Rv2783c* in PZA resistance. In our previous study, we identified four *M. tuberculosis* proteins (Rv2731, Rv2783, Rv3169, and Rv3601c) that bind to POA [6]; however, the role of these proteins in PZA action and resistance was not properly addressed. Recently, Rv2783 was shown to be involved in PZA resistance [19], where Asp67 of Rv2783 is indispensable for POA binding, but is inessential and adaptable for catalytic activity. POA binds to the Rv2783 protein, and its enzymatic activity is consequently obstructed by conformational changes. The D67N mutation inhibits POA binding to the mutant protein, so the mutant protein fulfills its enzymatic activities [19]. In this study, we discovered another mutation (A2104C) in *Rv2783c* from two clinical PZA-resistant strains lacking mutations in genes already reported to confer PZA resistance. Over-expression of *Rv2783c_A2104C_* significantly increased the PZA MIC in wild-type *M. tuberculosis*, showing straight evidence of the participation of *Rv2783c* in PZA resistance. In addition, we observed that both the Rv2783 and mutant Rv2783_K702Q_ could degrade the poly(A) using PNPase activity, while POA could only inhibit the degradative ability of wild-type Rv2783, not the mutant Rv2783_K702Q_. Furthermore, this novel mutation site is in the motif niche that is essential for the binding of both POA and tmRNA. Therefore, we evaluated whether POA competes with tmRNA to bind wild-type Rv2783 or Rv2783_K702Q_ and found that both Rv2783 and Rv2783_K702Q_ could be bound by tmRNA, although the mutant appeared to bind more weakly, while POA suppressed the binding ability of Rv2783, but not Rv2783_K702Q_. Taken together, we propose that POA and tmRNA compete for binding to the K702 site in Rv2783, except for the KXXXFXXF motif [7], but the Rv2783_K702Q_ mutant retains the ability to bind tmRNA as the mutant protein had reduced POA binding.

PNPase is found to bind tmRNA in some species, such as *Streptomyces aureofaciens* [21], which keeps the same KXXXFXXF motif for tmRNA binding as *M. tuberculosis*. This indicates that Rv2783 might be involved in regulating tmRNA-related trans-translation by specifically binding to tmRNA in *M. tuberculosis*. The very question of whether the binding of tmRNA to Rv2783c is essential or not for *M. tuberculosis* is not clear but can be addressed in the future.

The Rv2783 plays multiple important roles such as the PNPase enzyme, which exhibits functions in RNA synthesis and degradation [22] and is involved in persister formation in *E. coli* (https://www.biorxiv.org/content/10.1101/310987v1). As no transposon insertion within the *Rv2783c* gene was discovered in saturated *M. tuberculosis* transposon mutant libraries, this finding indicates the essentiality of *Rv2783c* for *M. tuberculosis* survival [23,24]. That POA inhibits multiple targets such as trans-translation (RpsA), energy production (PanD) and RNA degradation (PNPase) which are persistence pathways provides an explanation for its unique ability to kill persisters and shorten TB therapy. Improved understanding of PZA mode of action will help to design more effective therapies in the future.

In conclusion, we show that the K702Q mutation in Rv2783 is dispensable and flexible for the enzymatic activity of degrading poly(A) and binding to tmRNA, but is essential for POA binding. The Rv2783 protein binds to tmRNA, and this kind of binding ability could be obstructed by POA. POA binds to the Rv2783 protein, and hence, the convertible changes inhibit its enzymatic activities. The K702Q mutation in Rv2783 inhibits POA binding to the protein, and the ensuing mutant protein still maintains its catalytic activities. Nonetheless, functional analysis of this specific amino acid residue (Lys702) and further crystal structure analysis of Rv2783 are required to validate our hypothesis and present its role.

## 4. Materials and Methods

### 4.1. Bacterial Strains

The *M. tuberculosis* H37Ra strain was grown as previously described [4] and used for the construction of recombinant strains expressing target genes. Ninety-eight XDR-TB clinical isolates without *pncA* and *rpsA* mutations, including 56 PZA-resistant strains (MIC > 100 μg/mL) and 42 PZA-sensitive (MIC < 100 μg/mL) strains, were used for detection of possible *Rv2783c* mutations [25].

### 4.2. PZA Susceptibility Testing

Phenotypic PZA susceptibility testing was performed by the BACTEC MGIT 960 system with MGIT 960 PZA kits (BD Biosciences) following the manufacturer’s instructions but with a reduced inoculum of 0.25 mL of cell suspension [26]. The critical concentration of PZA used in this method was 100 μg/mL, and the BCG and H37Rv laboratory strains were used as PZA-resistant and PZA-susceptible controls, respectively.

### 4.3. PZase Activity Determination

The PZase biochemical assay was conducted as described by Wayne [27], with the following modifications from Zhang et al. [16,17]. Briefly, PZA in 100 µg/mL final concentration was used in 1 mL *M. tuberculosis* log phase cultures, with incubation at 37 °C overnight. The color development was read by adding 2% ferrous sulfate. A brown colored POA-Fe compound was considered positive for the PZase enzyme, which is from the reaction of POA converted from PZA with ferrous ions.

### 4.4. DNA Isolation, PCR, and DNA Sequencing of Rv2783c

Genomic DNA from 60 *M. tuberculosis* XDR-TB clinical isolates was isolated and a 2259 bp fragment including the whole *Rv2783c* gene was then amplified by PCR using primers A and B (Table 4). DNA sequence analysis was then performed according to standard methods to identify possible *Rv2783c* mutations as described elsewhere [16].

### 4.5. Rv2783c and Rv2783c_A2104C_ Mutant Gene Expression in M. tuberculosis

*Rv2873c* with its individual promoter regions was amplified by PCR from the *M. tuberculosis* H37Rv genome using primers C and D (Table 4) and then cloned into the pOLYG mycobacteria-*E. coli* shuttle vector (hygromycin resistant), which includes a strong *hsp60* promoter [28,29,30]. The empty vector and recombinant plasmids were electroporated into *M. tuberculosis* H37Ra cells as described previously [31]. Their effects on conferring possible drug resistance were then determined.

Site-directed mutagenesis of *Rv2783c* was fulfilled by primers E and F (Table 4) using the QuikChange™ site-directed mutagenesis kit, as described by the manufacturer (Stratagene). The mutant *Rv2783c_A2104C_* plasmid was constructed using the recombinant pOLYG vector with wild-type *Rv2783c* as a template and was introduced into *M. tuberculosis* H37Ra by electroporation, as described previously [31]. The effect of the mutant site on conferring PZA resistance was then tested as described below.

### 4.6. qRT-PCR Verification of Expression of Rv2783c Gene

The total RNA in the *M. tuberculosis* H37Ra strain with an empty vector and in the recombinant strains expressing the wild-type or mutant *Rv2783c* gene was isolated from cultures grown to the exponential phase, and the *Rv2783c* mRNA level was evaluated by qRT-PCR with primers G and H (Table 4). The fold change of its mRNA level to the level in the parental control *M. tuberculosis* H37Ra strain was determined after normalization to the *rrs* RNA (16S rRNA gene) level using primers I and J (Table 4), as described previously [32]. qRT-PCR was conducted in duplicate. The delta-delta Ct method was used for quantification [33].

### 4.7. Drug Susceptibility Testing (DST)

DST was performed for different *M. tuberculosis* H37Ra strains (wild-type, wild-type/pOLYG, wild-type/pOLYG*+Rv2783c*, and wild-type/pOLYG+*Rv2783c_A2104C_*) in 7H9 liquid medium for pyrazinamide (PZA) at acid pH 5.5 and for control drugs isoniazid (INH) and kanamycin (Kan) at pH 6.8 as described [31]. The following concentrations (μg/mL) using the proportion method were added: 0.03–0.6 for INH, 0.5–8.0 for KAN, and 50–1000 for PZA. Quality control was routinely performed during DST using the H37Rv and BCG reference strain.

The minimum inhibitory concentrations (MICs) (lowest concentration that prevents visible growth) of INH, Kan, and PZA in 7H9 broth were determined by a colorimetric method using 3-(4,5-dimethylthiazol-2-yl)-2,5-diphenyl tetrazolium bromide (MTT) as described [34].

### 4.8. Cloning, Sequencing, Expression, and Purification of M. tuberculosis Rv2783

The *Rv2783c* gene encoding the *M. tuberculosis* bifunctional guanosine pentaphosphate synthetase/polyribonucleotide nucleotidyltransferase was amplified by PCR from *M. tuberculosis* H37Rv using primers K and L. The *Rv2783c* PCR fragments were digested with NheI and HindIII and ligated to plasmid pET28a digested with the same enzymes to yield recombinant plasmids. The recombinant pET28a plasmid with wild-type *Rv2783c* was used as a template via site-directed mutagenesis, as mentioned before to obtain mutant plasmids. The mycobacterial Rv2783 and mutant Rv2783_K702Q_ proteins were expressed in *E. coli* strain BL21(DE3) after induction with IPTG (0.5 mM). Supernatant containing recombinant Rv2783 was purified on Ni^2+^-NTA agarose (Qiagen). Immobilized recombinant proteins were washed by a 20–50 mM imidazole gradient and eluted with buffer (20 mM Tris-HCl, 300 mM NaCl, pH 8.0 and 100–200 mM imidazole). The purified proteins were dialyzed against 10 mM Tris-HCl buffer (pH 7.5) to remove imidazole.

### 4.9. In Vitro Enzyme Inhibition

Rv2783 protein, which is a PNPase, as an ADP-producing enzyme, can be detected by ADP-Glo^TM^ (Promega—USA #V9101) [35], a homogeneous luminescence-based ADP detection assay. The assay is performed in two steps: First, ADP is produced via the PNPase reaction; second, ADP is converted into ATP that is detected via a luciferase/luciferin reaction. The luminescent signal is positively proportional to the ADP produced and, therefore, the PNPase activity. According to the protocol, 1 mM PolyA, 1× PBS, and the recombinant protein in different concentrations (0.01–0.5 µM) were incubated to start the phosphorylation reaction, and then an equal volume of Kinase Detection Reagent was added to simultaneously convert ADP to ATP and allow the newly synthesized ATP to be measured using a luciferase/luciferin reaction. The generated light is indicative of the PNPase activity. In the enzymatic experiment, the starting timepoint (0 min) in Figure 2b,c is when the Kinase Detection Reagent was simultaneously added to the reaction mixture.

To evaluate the ability of POA to inhibit the PNPase activity of Rv2783, first, POA in different concentrations (1–4 mM) and the recombinant protein were incubated for 10 min; then, 1 mM PolyA and 1× PBS were added into the reaction system and mixed fully. Finally, the reaction was incubated for 30 min with an equal volume of Kinase Detection Reagent (Promega) and read on a Biotek Synergy 2 multiplate reader. A reduced related light units (RLU) value is indicative of the inhibition ability of POA to PNPase. In the enzyme inhibition experiment, the starting timepoint (0 min) in Figure 2d,e is when Kinase Detection Reagent was added to the reaction mixture for 30 min incubation. The inhibition rate was calculated by the ratio of [RLU_inhibitor_ (15 min) − RLU_inhibitor_ (0 min)]/[RLU_control_(15 min) − RLU_control_ (0 min)].

### 4.10. Synthesis and Purification of tmRNA

The *M. tuberculosis ssrA* (tmRNA) gene under the control of the T7 promoter was amplified by PCR from *M. tuberculosis* H37Rv genomic DNA with primers M containing the T7 promoter sequence and N (Table 4). The *M. tuberculosis* tmRNA was transcribed in vitro and purified according to the manufacturer’s protocol of the Transcript Aid^TM^ T7 High Yield Transcription Kit (Fermentas).

### 4.11. RNA Electrophoretic Mobility Shift Assay (REMSA)

RNA electrophoretic mobility shift assay (REMSA) reactions [36] were conducted with a LightShift™ chemiluminescent REMSA kit (Thermo) with 1× REMSA binding buffer, 2 nM of biotin-labeled target RNA, 2 μM unlabeled target RNA, and a system-dependent Rv2783 or Rv2783_K702Q_ protein with or without POA in a 20 μL volume binding reaction per sample. REMSA reactions were carried out at room temperature for 30 min, and 5 μL of 5× Loading Buffer was then added to each 20 μL binding reaction, pipetting up and down several times to mix, prior to separation on a 6% polyacrylamide gel. The gel was pre-run at 100 V for 30 min and subsequently run at 100 V for 1 h after sample loading in 0.5× TBE buffer (89 mM Tris-acetate, 89 mM boric acid, 2 mM EDTA). The reaction products were then transferred to an Immobilon-NC membrane (Millipore Corp, Bedford, MA) at 100 V for 1 h and fixed by UV cross-linking. The biotin-labeled RNA was detected with a LightShift chemiluminescence REMSA kit (Pierce).

## Figures and Tables

**Figure 1 pathogens-08-00230-f001:**
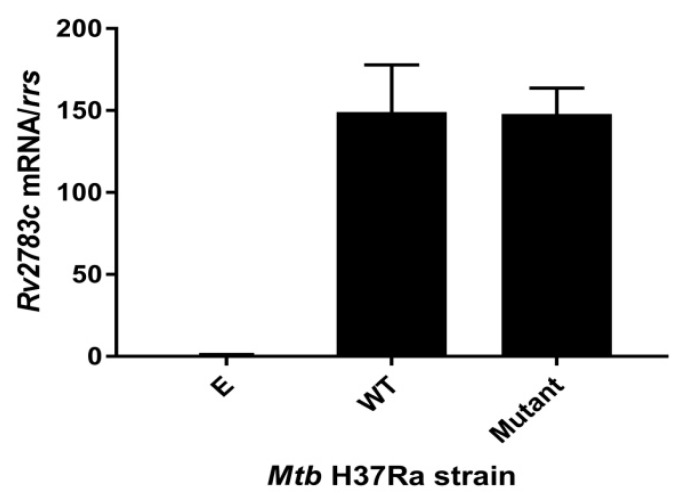
*Rv2783c* mRNA levels in the expression strains. Wild-type *Rv2783c* and *Rv2783c A2104C* mutants were expressed in the *M. tuberculosis* H37Ra strain, using the plasmid pOLYG. The fold increase is expressed as mRNA levels after normalizing to *rrs* mRNA values and then comparing to the parental *M. tuberculosis* control. Abbreviations are as follows: “E” = pOLYG, “WT” = pOLYG+*Rv2783c*, “Mutant” = pOLYG+*Rv2783c_A2104C_*.

**Figure 2 pathogens-08-00230-f002:**
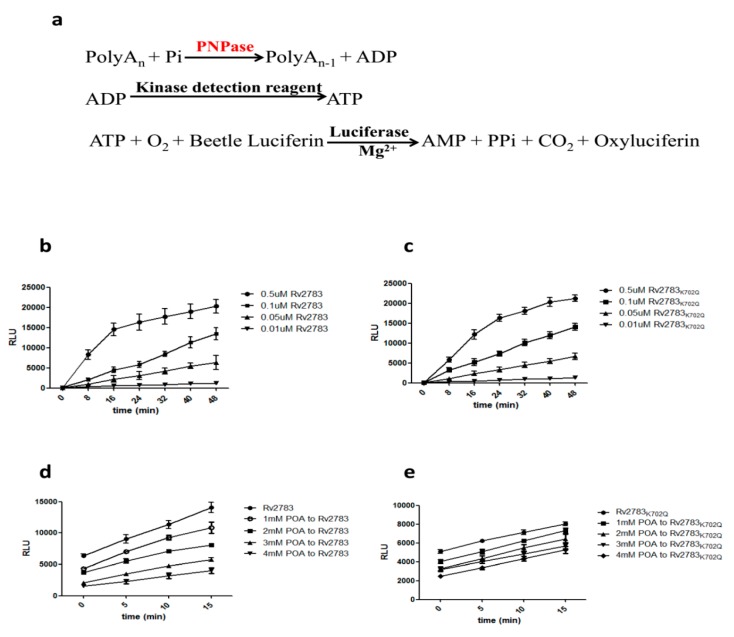
Effect of pyrazinoic acid (POA) on the RNA degradative ability of Rv2783. Rv2783 plays a major role in the degradation of RNA from the 3′- to 5′-ends in *M. tuberculosis*. (**a**) Coupled ADP-Glo™ assay protocol (adapted from Promega Corporation). The ADP-Glo™ Kinase Assay is a luminescent ADP detection assay that can measure the kinase activity of polyribonucleotide nucleotidyltransferase (PNPase) by quantifying the amount of ADP produced during a kinase reaction; (**b**) PNPase ability of Rv2783 is Rv2783 concentration-dependent; (**c**) PNPase ability of Rv2783_K702Q_ is also protein concentration-dependent; (**d**) inhibition of POA on the PNPase activity of WT Rv2783 is POA concentration-dependent; (**e**) no significant effect of POA on the PNPase activity of Rv2783_K702Q._ RLU = relative light units.

**Figure 3 pathogens-08-00230-f003:**
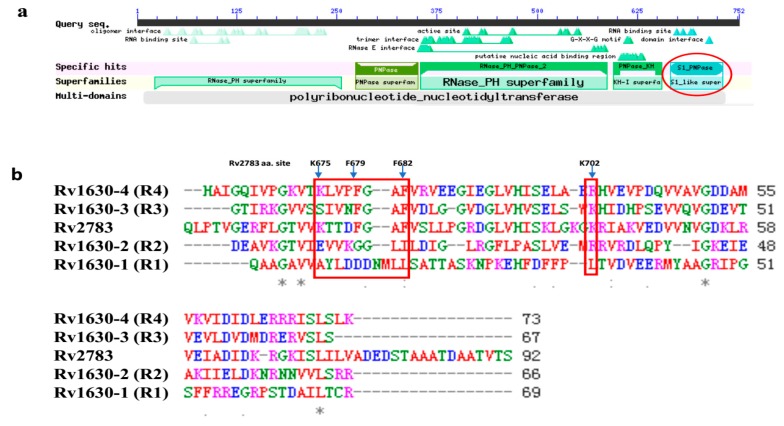
Rv2783 domain analysis and alignment with RpsA domains. (**a**) Rv2783 domain analysis shows the S1_PNPase domain (in the red circle) at its C-terminus, originally identified in ribosomal protein S1 (Rv1630) for RNA binding; (**b**) alignment of Rv2783 with four Rv1630 domains. Rv1630-1 to Rv1630-4 represent the four homologous RNA-binding domains (R1–R4) in RpsA. Both the Rv2783 S1_domain and the Rv1630-4 domain carry the KXXXFXXF motif (in the left red rectangle), which has been reported to comprise the main residues for POA and transfer-messenger RNA (tmRNA) binding. The mutant site of K702Q in Rv2783 is also located in the S1_domain (in the right red rectangle).

**Figure 4 pathogens-08-00230-f004:**
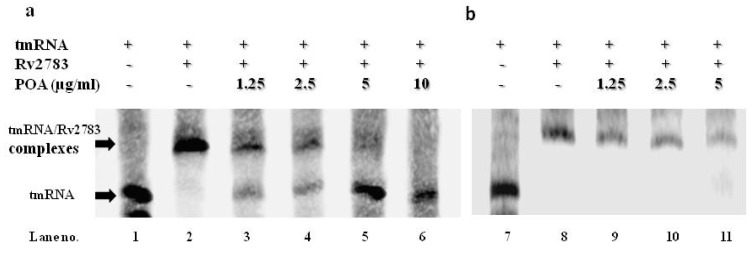
Effect of POA on the tmRNA binding ability of Rv2783. (**a**) Concentration-dependent inhibition of tmRNA binding to WT *M. tuberculosis (Mtb)* Rv2783 by POA (Lanes 2–7). tmRNA from *Mtb* was used as an RNA alone control (Lane 1); (**b**) no effect of tmRNA binding to *Mtb* Rv2783_K702Q_ by POA (Lanes 8–11). tmRNA had impaired binding to Rv2783_K702Q_ (Lane 8), and POA at different concentrations did not inhibit the interaction between Rv2783_K702Q_ and tmRNA (Lanes 9–11).

**Table 1 pathogens-08-00230-t001:** *Rv2783c* mutations associated with pyrazinamide (PZA) resistance in XDR-TB clinical isolates without *pncA* and *rpsA* mutations

Genotype in *Rv2783c*		PZA-Resistant ^2^			PZA-Susceptible		
NT ^1^ mutations	AA ^1^ mutations	n	#	%	n	#	%
A2104C	K702Q	56	2	4	42	0	0

^1^ NT: Nucleotide; AA: Amino acid. ^2^ Growth at higher than 100 mg/L PZA (pH 5.5) is defined as resistant to PZA.

**Table 2 pathogens-08-00230-t002:** Minimal inhibitory concentration (MIC) ^1^ values of *M. tuberculosis* strains expressing *Rv2783c* genes by the MTT liquid DST method ^2^.

Strain	PZA (pH 5.5)	INH	Kan
H37Ra	100	0.06	2
H37Ra/pOLYG	100	0.06	2
H37Ra/pOLYG+*Rv2783c*	200	0.06	2
H37Ra/pOLYG+*Rv2783c_A2104C_*	600	0.06	2

^1^ MICs were determined in duplicate at least three times by the MTT method. ^2^ DST was carried out to determine the effect of mutant *Rv2783c* expression on PZA resistance in *M. tuberculosis*.

**Table 3 pathogens-08-00230-t003:** Inhibition rate calculated by the ratio of [RLU_inhibitor_ (15 min) − RLU_inhibitor_ (0 min)]/[RLU_control_ (15 min) − RLU_control_ (0 min)]

Inhibitor	Rv2783	Rv2783_K702Q_
1mM POA	0.85	1.03
2mM POA	0.56	1.00
3mM POA	0.48	0.85
4mM POA	0.32	0.94

**Table 4 pathogens-08-00230-t004:** Primers used for generating *M. tuberculosis Rv2783c* overexpression, sequencing analysis, and site-directed mutagenesis of *Rv2783c* gene.

Primer	Sequence
Primers for PCR and DNA sequencing of *Rv2783c*	
A (F^1^)	5′-GCGTCACAGTCGGAAACCTAG-3′
B (R^2^)	5′-GTGCTCGGCTACACCAGGAC-3′
Primers for construction of *Rv2783c* expression in *Mtb* H37Ra strain	
C (F)	5′-GCTCTAGAATGTCTGCCGCTGAAATTGAC-3′(*Xba*I^3^)
D (R)	5′-CCAAGCTTTCAGCTGGTGACCGTCGCGGCATCG-3′(*Hind*III)
Primers for site-directed mutagenesis	
E (*K702Q*-F)	5′-GACTAGCAGCGGGTGGAGATCG-3′
F (*K702Q*-R)	5′-GCAGCTTGTCACCGACATTGACAAC-3′
Primers for qRT-PCR	
G (RT-F)	5′-CTACAACTTCCCGCCGTTCT-3′
H (RT-R)	5′-ATACGGGAATTCCTCGACGC-3′
I (rrs-F)	5′-GCGATACGGGCAGACTAGAG-3′
J (rrs-R)	5′-AAGGAAGGAAACCCACACCT-3′
Primers for construction of *Rv2783c* expression *E. coli* strains	
K (F)	5′-GGCTAGC ATGTCTGCCGCTGAAATTGAC-3′(*Nhe*I)
L (R)	5′-CCAAGCTTTCAGCTGGTGACCGTCGCGGCATCG-3′(*Hind*III)
Primers for PCR amplication of tmRNA	
M (L)	5′-TAATACGACTCACTATAGGATCTGACCGGGAAGTTAATGGC-3′
N (R)	5′-GATCAGATCCGGACGATCGGCATCG-3′

^1^ F = Forward primer ^2^ R = Reverse primer ^3^ Restriction sites incorporated into primers for cloning into pOLYG are underlined.

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
