# Peer review of "Pyrazinoic Acid Inhibits the Bifunctional Enzyme (Rv2783) in Mycobacterium tuberculosis by Competing with tmRNA"

_pathogens, 2019, doi:10.3390/pathogens8040230_

Round 1

Reviewer 1 Report

The authors report a point mutation in Rv2783 that is proposed to confer resistance to pyrazinamide (PZA). This point mutation was identified from 2 of 56 clinical isolates that are resistant to PZA. PZA already has a high MIC value against Mtb when given as a single agent (100 ug/mL), making the data  difficult to interpret. This is particularly true since Mtb H37Ra (wild type strain) is at 100 ug/ml, which is considered the breakpoint for resistance as defined by the authors. Overexpression of the mutant Rv2783 apparently raises the MIC to 600 ug/ml. 

In general, I differ in my opinion with the interpretation of the data and therefore the conclusions put forth by the authors.

mRNA levels are 150-fold higher compared to empty vector, which is dramatically higher than the change in PZA resistance. The change in the catalytic activity of Rv2783 is not very significant because such a high concentration of POA (metabolic product of PZA) is needed to inhibit. Based on the table 3, the IC50 against wild-type Rv2783 is around 2.5 mM, which is around 300 mg/L. This would suggest a substantially higher concentration would be needed to completely inhibit the enzyme and have any effect on MIC. The data in Figure 4 showing Rv2783 binding tmRNA are not clear. Firstly in part b, it looks like the gel was cut off since the band for tmRNA is not shown. Secondly, the band density corresponding to tmRNA/Rv2783 complex looks almost the same for the wild type (a) and mutant (b). The authors conclude that POA suppresses the binding of tmRNA to Rv2783. This seems to be the case with the wild type, since adding more POA makes the complex band more faint and the tmRNA band more dense. The density of the complex bands at different POA concentration for the mutant looks very similar to the wild-type complex Furthermore, the band for tmRNA is gone. How can you conclude that POA suppresses the binding affinity for wild-type but not the mutant? If so, what happens to the tmRNA. This data needs more rigor to support the conclusions.

Author Response

Comment: The authors report a point mutation in Rv2783 that is proposed to confer resistance to pyrazinamide (PZA). This point mutation was identified from 2 of 56 clinical isolates that are resistant to PZA. PZA already has a high MIC value against Mtb when given as a single agent (100 ug/mL), making the data  difficult to interpret. This is particularly true since Mtb H37Ra (wild type strain) is at 100 ug/ml, which is considered the breakpoint for resistance as defined by the authors. Overexpression of the mutant Rv2783 apparently raises the MIC to 600 ug/ml. 

Response: Thanks for the reviewer’s kind comment. PZA has already been verified to have a high MIC value against Mtb when given as a single agent (50-100 ug/mL) in some papers as belows.

Shi W, Zhang X, Jiang X, Yuan H, Lee JS, Barry CE, 3rd, Wang H, Zhang W, Zhang Y. Pyrazinamide inhibits trans-translation in Mycobacterium tuberculosis. Science. 2011;333:1630–1632.

Njire M,Wang N, Wang B, Tan Y, Cai X, Liu Y, Mugweru J, Guo J, Hameed HMA, Tan S, Liu J, Yew WW,Nuermberger E, Lamichhane G, Zhang T. 2017. Pyrazinoic acid inhibits a bifunctional enzyme in Mycobacterium tuberculosis. Antimicrob Agents Chemother 61:e00070-17. 

Xia Q, Zhao L, Li F, Fan Y, Chen Y, Wu B, et al. Phenotypic and genotypic characterization of pyrazinamide resistance among multidrug-resistant Mycobacterium tuberculosis isolates in Zhejiang, China. Antimicrob Agents Chemother. 2015;59:1690–1695.

Comment: In general, I differ in my opinion with the interpretation of the data and therefore the conclusions put forth by the authors.

mRNA levels are 150-fold higher compared to empty vector, which is dramatically higher than the change in PZA resistance. The change in the catalytic activity of Rv2783 is not very significant because such a high concentration of POA (metabolic product of PZA) is needed to inhibit. Based on the table 3, the IC50 against wild-type Rv2783 is around 2.5 mM, which is around 300 mg/L. This would suggest a substantially higher concentration would be needed to completely inhibit the enzyme and have any effect on MIC.

Response: Thanks for the reviewer’s suggestion. As mRNA is eventually translated into protein, it is usually assumed that there is some sort of correlation between the levels of mRNA and protein. But, there are presumably reasons for the poor correlations generally reported between the level of mRNA and the level of protein, and these may not be mutually exclusive. First, there are many complicated and varied post-transcriptional mechanisms involved in turning mRNA into protein that are not yet sufficiently well defined to be able to compute protein concentrations from mRNA; second, proteins may differ substantially in their in vivo half lives; and/or third, there is a significant amount of error and noise in both protein and mRNA experiments that limit our ability to get a clear picture. However, we found that a substantially higher concentration of POA was needed to completely inhibit the enzyme of Rv2783 and have any effect on MIC based on our experiment.

Comment: The data in Figure 4 showing Rv2783 binding tmRNA are not clear. Firstly in part b, it looks like the gel was cut off since the band for tmRNA is not shown. Secondly, the band density corresponding to tmRNA/Rv2783 complex looks almost the same for the wild type (a) and mutant (b). The authors conclude that POA suppresses the binding of tmRNA to Rv2783. This seems to be the case with the wild type, since adding more POA makes the complex band more faint and the tmRNA band more dense. The density of the complex bands at different POA concentration for the mutant looks very similar to the wild-type complex Furthermore, the band for tmRNA is gone. How can you conclude that POA suppresses the binding affinity for wild-type but not the mutant? If so, what happens to the tmRNA. This data needs more rigor to support the conclusions.

Response: We are grateful for the reviewer’s suggestion. We are sorry for the ambiguity of Figure 4b. We lowered the contrast ratio of Figure 4 to show the bands more clearly and we also provided the original figures of EMSA (Supporting Figure 3 and Figure 4) to show the phenomenon and avoid the ambiguity in the re-submitted manuscript. According to the supporting Figure, the mutant Rv2783K702Q and the binding ability of the mutant Rv2783K702Q was tmRNA was not hampered when POA was added, therefore the band of tmRNA was not gone.

Reviewer 2 Report

This study showed that the bifunctional enzyme, Rv2783 was associated with PZA resistance in clinical isolates. They identified a new mutation in Rv2783 which contribute to the PZA resistance. It is important that they find the mutation contributes to the PZA resistance through binding with tmRNA without any affection by the POA. However, there are some issues need to be addressed in the current version.

First, the author algins the sequence from 56-PZA-resistant clinical isolates. The mutation A2104C is only identified in two clinical isolates. Does that mean this mutation is not conserved across the different clinical strains? Also, maybe this mutation is randomly generated and minimally contributes to the PZA-resistance. Did the author find any other mutations in other proteins?  Please provide more information about the Table 2. Like the INH and Kan. Although the author found that this mutation did not change the mRNA level of Rv2783, has the author verified whether the mutation affects the protein level in TB? Revise the figure 2a .  Please provide the Coomassie Blue stain gel or WB to show the purified protein Rv2783 and the Rv2783K702Q. For the EMSA assay, figure b is not a nice picture. Could you provide a better picture to show this?

Author Response

Comment: This study showed that the bifunctional enzyme, Rv2783 was associated with PZA resistance in clinical isolates. They identified a new mutation in Rv2783 which contribute to the PZA resistance. It is important that they find the mutation contributes to the PZA resistance through binding with tmRNA without any affection by the POA. However, there are some issues need to be addressed in the current version.

First, the author algins the sequence from 56-PZA-resistant clinical isolates. The mutation A2104C is only identified in two clinical isolates. Does that mean this mutation is not conserved across the different clinical strains?  Also, maybe this mutation is randomly generated and minimally contributes to the PZA-resistance. Did the author find any other mutations in other proteins?  

Response: We are thankful for the reviewer’s comment. Firstly, we did not perform Whole Genome Sequencing for the two isolates that had the Rv2783c mutation. It was identified by targeted Sanger sequencing of Rv2783c from a panel of 56 PZA-resistant clinical isolates while no pncA, rpsA and panD mutations which play roles in PZA resistance were found. Therefore there might be other mutations in other proteins. Furthermore, we provided the possibility between the Rv2783cK702Q mutation and PZA resistance from the selection of clinical strains here and then used the following assay to verify the correlation and seek for the possible mechanism.

Comment: Please provide more information about the Table 2. Like the INH and Kan. 

Response: Thanks for the reviewer’s kind remind. We have already complemented the description of results about Table 2 in the re-submitted manuscript (Line88-90).

“Introduction of the K702Q point mutation into M. tuberculosis H37Ra strain did not alter the susceptibility to other drugs including INH and Kan.”

Comment: Although the author found that this mutation did not change the mRNA level of Rv2783, has the author verified whether the mutation affects the protein level in TB? 

Response: We are grateful for the reviewer’s comment. We are sorry for that we didn’t verified whether the mutation affects the protein level in TB since we are short of the antibody of the Rv2783 protein which we are trying to obtain. We will try to verify protein level in TB once we get the antibody.

Comment: Revise the figure 2a .  

Response: Thanks for the reviewer’s suggestion. We have already corrected Figure 2a in the re-submitted manuscript.

Comment: Please provide the Coomassie Blue stain gel or WB to show the purified protein Rv2783 and the Rv2783K702Q.

Response: Thanks for the reviewer’s comment. We have already provided the Coomassie Blue stain gel to show the purified protein Rv2783 and the Rv2783K702Q as Supporting Figure 1 and Figure 2 in the re-submitted manuscript.

Comment: For the EMSA assay, figure b is not a nice picture. Could you provide a better picture to show this?

Response: We are grateful for the reviewer’s suggestion. We are sorry for the ambiguity of Figure 4b. We lowered the contrast ratio of this Figure to show the bands more clearly and we also provided the original figures of EMSA (Supporting Figure 3 and Figure 4) to show the phenomenon and avoid the ambiguity in the re-submitted manuscript.

Reviewer 3 Report

Figure 2a requires correction.

Author Response

Comment: Figure 2a requires correction.

Response: We are thankful for the reviewer’s kind comment. We have already corrected Figure 2a in the re-submitted manuscript.

Round 2

Reviewer 1 Report

All comments were adequately addressed

Reviewer 2 Report

The author addressed most of my concerns; I agree to accept this paper.